# FIDELITY-WEIGHTED LEARNING

**Mostafa Dehghani**
University of Amsterdam
dehghani@uva.nl

**Arash Mehrjou**
MPI for Intelligent Systems
amehrjou@tuebingen.mpg.de

**Stephan Gouws**
Google Brain
sgouws@google.com

**Jaap Kamps**
University of Amsterdam
kamps@uva.nl

**Bernhard Schölkopf**
MPI for Intelligent Systems
bs@tuebingen.mpg.de

## ABSTRACT

Training deep neural networks requires many training samples, but in practice training labels are expensive to obtain and may be of varying quality, as some may be from trusted expert labelers while others might be from heuristics or other sources of weak supervision such as crowd-sourcing. This creates a fundamental quality-versus-quantity trade-off in the learning process. Do we learn from the small amount of high-quality data or the potentially large amount of weakly-labeled data? We argue that if the learner could somehow know and take the label-quality into account when learning the data representation, we could get the best of both worlds. To this end, we propose "fidelity-weighted learning" (FWL), a semi-supervised student-teacher approach for training deep neural networks using weakly-labeled data. FWL modulates the parameter updates to a *student* network (trained on the task we care about) on a per-sample basis according to the posterior confidence of its label-quality estimated by a *teacher* (who has access to the high-quality labels). Both student and teacher are learned from the data. We evaluate FWL on two tasks in information retrieval and natural language processing where we outperform state-of-the-art alternative semi-supervised methods, indicating that our approach makes better use of strong and weak labels, and leads to better task-dependent data representations.

## 1 INTRODUCTION

The success of deep neural networks to date depends strongly on the availability of labeled data which is costly and not always easy to obtain. Usually it is much easier to obtain small quantities of high-quality labeled data and large quantities of unlabeled data. The problem of how to best integrate these two different sources of information during training is an active pursuit in the field of semi-supervised learning (Chapelle et al., 2006). However, for a large class of tasks it is also easy to define one or more so-called "weak annotators", additional (albeit noisy) sources of *weak supervision* based on heuristics or "weaker", biased classifiers trained on e.g. non-expert crowd-sourced data or data from different domains that are related. While easy and cheap to generate, it is not immediately clear if and how these additional weakly-labeled data can be used to train a stronger classifier for the task we care about. More generally, in almost all practical applications machine learning systems have to deal with data samples of variable quality. For example, in a large dataset of images only a small fraction of samples may be labeled by experts and the rest may be crowd-sourced using e.g. Amazon Mechanical Turk (Veit et al., 2017). In addition, in some applications, labels are intentionally perturbed due to privacy issues (Wainwright et al., 2012; Papernot et al., 2017).

Assuming we can obtain a large set of weakly-labeled data in addition to a much smaller training set of "strong" labels, the simplest approach is to expand the training set by including the weakly-supervised samples (all samples are equal). Alternatively, one may pretrain on the weak data and then fine-tune on observations from the true function or distribution (which we call strong data). Indeed, it has recently been shown that a small amount of expert-labeled data can be augmented in such a way by a large set of raw data, with labels coming from a heuristic function, to train a more accurate neural ranking model (Dehghani et al., 2017d). The downside is that such approaches are oblivious to the amount or source of noise in the labels.

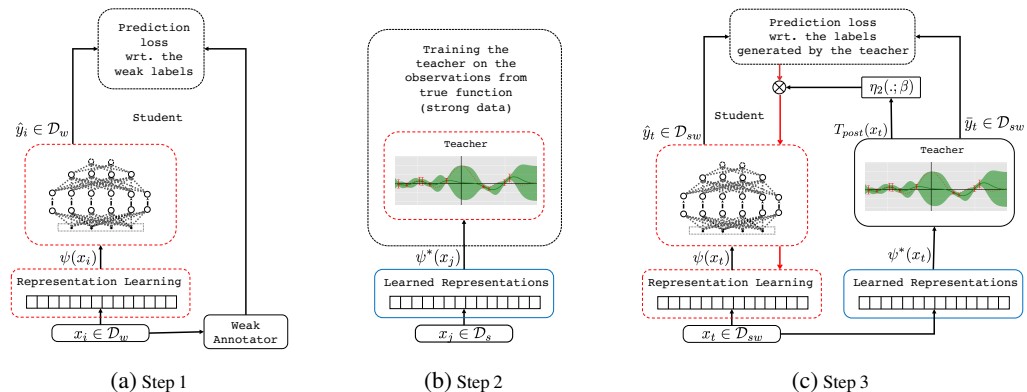

Figure 1: Illustration of Fidelity-Weighted Learning: Step 1: Pre-train student on weak data, Step 2: Fit teacher to observations from the true function, and Step 3: Fine-tune student on labels generated by teacher, taking the confidence into account. Red dotted borders and blue solid borders depict components with trainable and non-trainable parameters, respectively.

In this paper, we argue that treating weakly-labeled samples uniformly (i.e. each weak sample contributes equally to the final classifier) ignores potentially valuable information of the label quality. Instead, we propose Fidelity-Weighted Learning (FWL), a Bayesian semi-supervised approach that leverages a small amount of data with true labels to generate a larger training set with *confidence-weighted weakly-labeled samples*, which can then be used to modulate the fine-tuning process based on the fidelity (or quality) of each weak sample. By directly modeling the inaccuracies introduced by the weak annotator in this way, we can control the extent to which we make use of this additional source of weak supervision: more for confidently-labeled weak samples close to the true observed data, and less for uncertain samples further away from the observed data.

We propose a setting consisting of two main modules. One is called the student and is in charge of learning a suitable data representation and performing the main prediction task, the other is the teacher which modulates the learning process by modeling the inaccuracies in the labels. We explain our approach in much more detail in Section 2, but at a high level it works as follows (see Figure 1): We pretrain the student network on weak data to learn an initial task-dependent data representation which we pass to the teacher along with the strong data. The teacher then learns to predict the strong data, but crucially, *based on the student's learned representation*. This then allows the teacher to generate new labeled training data from unlabeled data, and in the process correct the student's mistakes, leading to a better final data representation and better final predictor.

We introduce the proposed FWL approach in more detail in Section 2. We then present our experimental setup in Section 3 where we evaluate FWL on a toy task and two real-world tasks, namely document ranking and sentence sentiment classification. In all cases, FWL outperforms competitive baselines and yields state-of-the-art results, indicating that FWL makes better use of the limited true labeled data and is thereby able to learn a better and more meaningful task-specific representation of the data. Section 4 provides analysis of the bias-variance trade-off and the learning rate, suggesting also to view FWL from the perspective of Vapnik's learning with privileged information (LUPI) framework (Vapnik & Izmailov, 2015). Section 5 situates FWL relative to related work, and we end the paper by drawing the main conclusions in Section 6.

## 2 FIDELITY-WEIGHTED LEARNING (FWL)

In this section, we describe our proposed FWL approach for semi-supervised learning when we have access to weak supervision (e.g. heuristics or weak annotators). We assume we are given a large set of unlabeled data samples, a heuristic labeling function called the *weak annotator*, and a small set of high-quality samples labeled by experts, called the *strong dataset*, consisting of tuples of training samples $x_i$ and their true labels $y_i$, i.e. $\mathcal{D}_s = \{(x_i, y_i)\}$. We consider the latter to be observations from the true target function that we are trying to learn. We use the weak annotator to generate labels for the unlabeled samples. Generated labels are noisy due to the limited accuracy of the weak annotator. This gives us the *weak dataset* consisting of tuples of training samples $x_i$ and their weak labels $\tilde{y}_i$, i.e. $\mathcal{D}_w = \{(x_i, \tilde{y}_i)\}$. Note that we can generate a large amount of weak training data $\mathcal{D}_w$ at almost no cost using the weak annotator. In contrast, we have only a limited amount of observations from the true function, i.e. $|\mathcal{D}_s| \ll |\mathcal{D}_w|$.

---

**Algorithm 1** Fidelity-Weighted Learning.

---

1: Train the student on samples from the weakly-annotated data $D_w$.

2: Freeze the representation-learning component $\psi(.)$ of the student and train teacher on the strong data $D_s = (\psi(x_j), y_j)$. Apply teacher to unlabeled samples $x_t$ to obtain soft dataset $D_{sw} = \{(x_t, \bar{y}_t)\}$ where $\bar{y}_t = T(x_t)$ is the soft label and for each instance $x_t$, the uncertainty of its label, $\Sigma(x_t)$, is provided by the teacher.

3: Train the student on samples from $D_{sw}$ with SGD and modulate the step-size $\eta_t$ according to the per-sample quality estimated using the teacher (Equation 1).

---

Our proposed setup comprises a neural network called the **student** and a Bayesian function approximator called the **teacher**. The training process consists of three phases which we summarize in Algorithm 1 and Figure 1.

**Step 1** *Pre-train the student on $\mathcal{D}_w$ using weak labels generated by the weak annotator.*

The main goal of this step is to learn a *task dependent* representation of the data as well as pretraining the student. The student function is a neural network consisting of two parts. The first part $\psi(.)$ learns the data representation and the second part $\phi(.)$ performs the prediction task (e.g. classification). Therefore the overall function is $\hat{y} = \phi(\psi(x_i))$. The student is trained on all samples of the weak dataset $\mathcal{D}_w = \{(x_i, \tilde{y}_i)\}$. For brevity, in the following, we will refer to both data sample $x_i$ and its representation $\psi(x_i)$ by $x_i$ when it is obvious from the context. From the self-supervised feature learning point of view, we can say that representation learning in this step is solving a surrogate task of approximating the expert knowledge, for which a noisy supervision signal is provided by the weak annotator.

**Step 2** *Train the teacher on the strong data $(\psi(x_j), y_j) \in \mathcal{D}_s$ represented in terms of the student representation $\psi(.)$ and then use the teacher to generate a soft dataset $\mathcal{D}_{sw}$ consisting of $\langle sample, predicted\ label, confidence\rangle$ for **all** data samples.*

We use a Gaussian process as the teacher to capture the label uncertainty in terms of the student representation, estimated w.r.t the strong data. We explain the finer details of the $\mathcal{GP}$ in Appendix C, and just present the overall description here. A prior mean and co-variance function is chosen for $\mathcal{GP}$. The learned embedding function $\psi(\cdot)$ in Step 1 is then used to map the data samples to dense vectors as input to the $\mathcal{GP}$. We use the learned representation by the student in the previous step to compensate lack of data in $\mathcal{D}_s$ and the teacher can enjoy the learned knowledge from the large quantity of the weakly annotated data. This way, we also let the teacher see the data through the lens of the student.

The $\mathcal{GP}$ is trained on the samples from $\mathcal{D}_s$ to learn the posterior mean $\boldsymbol{m}_{\text{post}}$ (used to generate soft labels) and posterior co-variance $K_{\text{post}}(., .)$ (which represents label uncertainty). We then create the *soft dataset* $\mathcal{D}_{sw} = \{(x_t, \bar{y}_t)\}$ using the posterior $\mathcal{GP}$, input samples $x_t$ from $\mathcal{D}_w \cup \mathcal{D}_s$, and predicted labels $\bar{y}_t$ with their associated uncertainties as computed by $T(x_t)$ and $\Sigma(x_t)$:

$$
\begin{aligned}
T(x_t) &= g(\boldsymbol{m}_{\text{post}}(x_t)) \\
\Sigma(x_t) &= h(K_{\text{post}}(x_t, x_t))
\end{aligned}
$$

The generated labels are called *soft labels*. Therefore, we refer to $\mathcal{D}_{sw}$ as a soft dataset. $g(.)$ transforms the output of $\mathcal{GP}$ to the suitable output space. For example in classification tasks, $g(.)$ would be the softmax function to produce probabilities that sum up to one. For multidimensional-output tasks where a vector of variances is provided by the $\mathcal{GP}$, the vector $K_{\text{post}}(x_t, x_t)$ is passed through an aggregating function $h(.)$ to generate a scalar value for the uncertainty of each sample. Note that we train $\mathcal{GP}$ only on the strong dataset $\mathcal{D}_s$ but then use it to generate soft labels $\bar{y}_t = T(x_t)$ and uncertainty $\Sigma(x_t)$ for samples belonging to $\mathcal{D}_{sw} = \mathcal{D}_w \cup \mathcal{D}_s$.

In practice, we furthermore divide the space of data into several regions and assign each region a separate $\mathcal{GP}$ trained on samples from that region. This leads to a better exploration of the data space and makes use of the inherent structure of data. The algorithm called clustered $\mathcal{GP}$ gave better results compared to a single GP. See Appendix A for the detailed description and empirical observations which makes the use of multiple $\mathcal{GP}$s reasonable.

**Step 3** *Fine-tune the weights of the student network on the soft dataset, while modulating the magnitude of each parameter update by the corresponding teacher-confidence in its label.*

The student network of Step 1 is fine-tuned using samples from the soft dataset $\mathcal{D}_{sw} = \{(x_t, \bar{y}_t)\}$ where $\bar{y}_t = T(x_t)$. The corresponding uncertainty $\Sigma(x_t)$ of each sample is mapped to a confidence value

according to Equation 1 below, and this is then used to determine the step size for each iteration of the stochastic gradient descent (SGD). So, intuitively, for data points where we have true labels, the uncertainty of the teacher is almost zero, which means we have high confidence and a large step-size for updating the parameters. However, for data points where the teacher is not confident, we down-weight the training steps of the student. This means that at these points, we keep the student function as it was trained on the weak data in Step 1.

More specifically, we update the parameters of the student by training on $\mathcal{D}_{sw}$ using SGD:

$$\boldsymbol{w}^* = \operatorname*{argmin}_{\boldsymbol{w} \in \mathcal{W}} \frac{1}{N} \sum_{(x_t, \bar{y}_t) \in \mathcal{D}_{sw}} l(\boldsymbol{w}, x_t, \bar{y}_t) + \mathcal{R}(\boldsymbol{w}),$$

$$\boldsymbol{w}_{t+1} = \boldsymbol{w}_t - \eta_t (\nabla l(\boldsymbol{w}, x_t, \bar{y}_t) + \nabla \mathcal{R}(\boldsymbol{w}))$$

where $l(\cdot)$ is the per-example loss, $\eta_t$ is the total learning rate, $N$ is the size of the soft dataset $\mathcal{D}_{sw}$, $\boldsymbol{w}$ is the parameters of the student network, and $\mathcal{R}(.)$ is the regularization term.

We define the total learning rate as $\eta_t = \eta_1(t)\eta_2(x_t)$, where $\eta_1(t)$ is the usual learning rate of our chosen optimization algorithm that anneals over training iterations, and $\eta_2(x_t)$ is a function of the label uncertainty $\Sigma(x_t)$ that is computed by the teacher for each data point. Multiplying these two terms gives us the total learning rate. In other words, $\eta_2$ represents the *fidelity* (quality) of the current sample, and is used to multiplicatively modulate $\eta_1$. Note that the first term does not necessarily depend on each data point, whereas the second term does. We propose

$$\eta_2(x_t) = \exp[-\beta \Sigma(x_t)], \tag{1}$$

to exponentially decrease the learning rate for data point $x_t$ if its corresponding soft label $\bar{y}_t$ is unreliable (far from a true sample). In Equation 1, $\beta$ is a positive scalar hyper-parameter. Intuitively, small $\beta$ results in a student which listens more carefully to the teacher and copies its knowledge, while a large $\beta$ makes the student pay less attention to the teacher, staying with its initial weak knowledge. More concretely speaking, as $\beta \to 0$ student places more trust in the labels $\bar{y}_t$ estimated by the teacher and the student copies the knowledge of the teacher. On the other hand, as $\beta \to \infty$, student puts less weight on the extrapolation ability of $\mathcal{GP}$ and the parameters of the student are not affected by the correcting information from the teacher.

## 3 EXPERIMENTS

In this section, we apply FWL first to a toy problem and then to two different real tasks: *document ranking* and *sentiment classification*. The neural networks are implemented in TensorFlow (Abadi et al., 2015; Tang, 2016). GPflow (Matthews et al., 2017) is employed for developing the $\mathcal{GP}$ modules. For both tasks, we evaluate the performance of our method compared to the following baselines:

1. **WA**. The weak annotator, i.e. the unsupervised method used for annotating the unlabeled data.
2. **NN$_{\mathbf{W}}$**. The student trained only on weak data.
3. **NN$_{\mathbf{S}}$**. The student trained only on strong data.
4. **NN$_{\mathbf{S}+/\mathbf{W}}$**. The student trained on samples that are alternately drawn from $\mathcal{D}_w$ without replacement, and $\mathcal{D}_s$ with replacement. Since $|\mathcal{D}_s| \ll |\mathcal{D}_w|$, it oversamples the strong data.
5. **NN$_{\mathbf{W} \to \mathbf{S}}$**. The student trained on weak dataset $\mathcal{D}_w$ and fine-tuned on strong dataset $\mathcal{D}_s$.
6. **NN$_{\mathbf{W}^\omega \to \mathbf{S}}$**. The student trained on the weak data, but the step-size of each weak sample is weighted by a fixed value $0 \le \omega \le 1$, and fine-tuned on strong data. As an approximation for the optimal value for $\omega$, we have used the mean of $\eta_2$ of our model (below).
7. **FWL** *unsuprep*. The representation in the first step is trained in an unsupervised way[1] and the student is trained on examples labeled by the teacher using the confidence scores.
8. **FWL** $\backslash \Sigma$. The student trained on the weakly labeled data and fine-tuned on examples labeled by the teacher without taking the confidence into account. This baseline is similar to (Veit et al., 2017).
9. **FWL**. Our FWL model, i.e. the student trained on the weakly labeled data and fine-tuned on examples labeled by the teacher using the confidence scores.

In the following, we introduce each task and the results produced for it, more detail about the exact student network and teacher $\mathcal{GP}$ for each task are in the appendix.

---

[1]In the document ranking task, as the representation of documents and queries, we use weighted averaging over pretrained embeddings of their words based on their inverse document frequency (Dehghani et al., 2017d). In the sentiment analysis task, we use skip-thoughts vectors(Kiros et al., 2015)

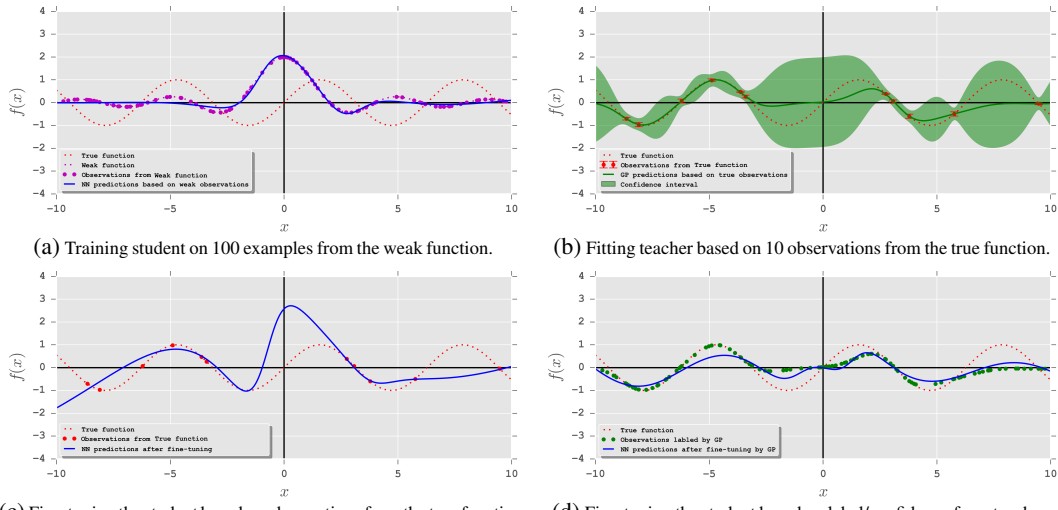

(a) Training student on 100 examples from the weak function.

(b) Fitting teacher based on 10 observations from the true function.

(c) Fine-tuning the student based on observations from the true function.

(d) Fine-tuning the student based on label/confidence from teacher.

Figure 2: Toy example: The true function we want to learn is $y = \sin(x)$ and the weak function is $y = 2sinc(x)$.

## 3.1 TOY PROBLEM

We first apply FWL to a one-dimensional toy problem to illustrate the various steps. Let $f_t(x) = \sin(x)$ be the true function (red dotted line in Figure 2a) from which a small set of observations $\mathcal{D}_s = \{x_j, y_j\}$ is provided (red points in Figure 2b). These observation might be noisy, in the same way that labels obtained from a human labeler could be noisy. A weak annotator function $f_w(x) = 2sinc(x)$ (magenta line in Figure 2a) is provided, as an approximation to $f_t(.)$.

The task is to obtain a good estimate of $f_t(.)$ given the set $\mathcal{D}_s$ of strong observations and the weak annotator function $f_w(.)$. We can easily obtain a large set of observations $\mathcal{D}_w = \{x_i, \tilde{y}_i\}$ from $f_w(.)$ with almost no cost (magenta points in Figure 2a).

We consider two experiments:

1. A neural network trained on weak data and then fine-tuned on strong data from the true function, which is the most common semi-supervised approach (Figure 2c).
2. A teacher-student framework working by the proposed FWL approach.

As can be seen in Figure 2d, FWL by taking into account label confidence, gives a better approximation of the true hidden function. We repeated the above experiment 10 times. The average RMSE with respect to the true function on a set of test points over those 10 experiments for the student, were as follows:

1. Student is trained on weak data (blue line in Figure 2a): 0.8406,
2. Student is trained on weak data then fine tuned on true observations (blue line in Figure 2c): 0.5451,
3. Student is trained on weak data, then fine tuned by soft labels and confidence information provided by the teacher (blue line in Figure 2d): 0.4143 (best).

More details of the neural network and $\mathcal{GP}$ along with the specification of the data used in the above experiment are presented in Appendix C and E.1.

## 3.2 DOCUMENT RANKING

This task is the core information retrieval problem and is challenging as the ranking model needs to learn a representation for long documents and capture the notion of relevance between queries and documents. Furthermore, the size of publicly available datasets with query-document relevance judgments is unfortunately quite small ($\sim 250$ queries). We employ a state-of-the-art pairwise neural ranker architecture as the student (Dehghani et al., 2017d). In this model, ranking is cast as a regression task. Given each training sample $x$ as a triple of query $q$, and two documents $d^+$ and $d^-$, the goal is to learn a function $\mathcal{F}: \{<q, d^+, d^->\} \to \mathbb{R}$, which maps each data sample $x$ to a scalar output value $y$ indicating the probability of $d^+$ being ranked higher than $d^-$ with respect to $q$.

Table 1: Performance of FWL approach and baseline methods for ranking task. $\blacktriangle^i$ indicates that the improvements with respect to the baseline $i$ are statistically significant at the 0.05 level using the paired two-tailed t-test with Bonferroni correction.

| | Method | Robust04 | | ClueWeb | |
|---|---|---|---|---|---|
| | | MAP | nDCG@20 | MAP | nDCG@20 |
| 1 | $WA_{BM25}$ | $0.2503^{\blacktriangle 37}$ | $0.4102^{\blacktriangle 37}$ | $0.1021^{\blacktriangle 37}$ | $0.2070^{\blacktriangle 37}$ |
| 2 | $NN_W$ (Dehghani et al., 2017d) | $0.2702^{\blacktriangle 137}$ | $0.4290^{\blacktriangle 137}$ | $0.1297^{\blacktriangle 137}$ | $0.2201^{\blacktriangle 137}$ |
| 3 | $NN_S$ | 0.1790 | 0.3519 | 0.0782 | 0.1730 |
| 4 | $NN_{S+/W}$ | $0.2763^{\blacktriangle 1237}$ | $0.4330^{\blacktriangle 1237}$ | $0.1354^{\blacktriangle 1237}$ | $0.2319^{\blacktriangle 1237}$ |
| 5 | $NN_{W \to S}$ | $0.2810^{\blacktriangle 1237}$ | $0.4372^{\blacktriangle 1237}$ | $0.1346^{\blacktriangle 1237}$ | $0.2317^{\blacktriangle 1237}$ |
| 6 | $NN_{W^\omega \to S}$ | $0.2899^{\blacktriangle 123457}$ | $0.4431^{\blacktriangle 123457}$ | $0.1320^{\blacktriangle 12347}$ | $0.2309^{\blacktriangle 12347}$ |
| 7 | $FWL_{unsuprep}$ | $0.2211^{\blacktriangle 37}$ | $0.3700^{\blacktriangle 37}$ | $0.0831^{\blacktriangle 37}$ | $0.1964^{\blacktriangle 37}$ |
| 8 | $FWL \setminus \Sigma$ | $0.2980^{\blacktriangle 123457}$ | $0.4516^{\blacktriangle 123457}$ | $0.1386^{\blacktriangle 123457}$ | $0.2340^{\blacktriangle 123457}$ |
| 9 | FWL | $\mathbf{0.3124}^{\blacktriangle 12345678}$ | $\mathbf{0.4607}^{\blacktriangle 12345678}$ | $\mathbf{0.1472}^{\blacktriangle 12345678}$ | $\mathbf{0.2453}^{\blacktriangle 12345678}$ |

**The student** follows the architecture proposed in (Dehghani et al., 2017d). The first layer of the network, i.e. representation learning layer $\psi : \{<q, d^+, d^->\} \to \mathbb{R}^m$ maps each input sample to an $m$- dimensional real-valued vector. In general, besides learning embeddings for words, function $\psi$ learns to compose word embedding based on their global importance in order to generate query/document embeddings. The representation layer is followed by a simple fully-connected feed-forward network with a sigmoidal output unit to predict the probability of ranking $d^+$ higher than $d^-$. The general schema of the student is illustrated in Figure 3. More details are provided in Appendix B.1.

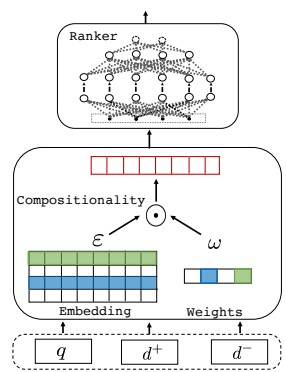

Figure 3: The student for the document ranking task.

**The teacher** is implemented by clustered $\mathcal{GP}$ algorithm. See Appendix C for more details.

**The weak annotator** is BM25 (Robertson & Zaragoza, 2009), a well-known unsupervised method for scoring query-document pairs based on statistics of the matched terms. More details are provided in Appendix D.1.

Description of the data with weak labels and data with true labels as well as the setup of the document-ranking experiments is presented in Appendix E.2 in more details.

**Results and Discussions** We conducted k-fold cross validation on $\mathcal{D}_s$ (the strong data) and report two standard evaluation metrics for ranking: mean average precision (MAP) of the top-ranked $1,000$ documents and normalized discounted cumulative gain calculated for the top 20 retrieved documents (nDCG@20). Table 1 shows the performance on both datasets. As can be seen, FWL provides a significant boost on the performance over all datasets. In the ranking task, the student is designed in particular to be trained on weak annotations (Dehghani et al., 2017d), hence training the network only on weak supervision, i.e. $NN_W$ performs better than $NN_S$. This can be due to the fact that ranking is a complex task requiring many training samples, while relatively few data with true labels are available.

Alternating between strong and weak data during training, i.e. $NN_{S+/W}$ seems to bring little (but statistically significant) improvement. However, we can gain better results by the typical fine-tuning strategy, $NN_{W \to S}$. Comparing the performance of $FWL_{unsuprep}$ to FWL indicates that, first of all learning the representation of the input data downstream of the main task leads to better results compared to a task-independent unsupervised or self-supervised way. Also the dramatic drop in the performance compared to the FWL, emphasizes the importance of the preretraining the student on weakly labeled data. We can gain improvement by fine-tuning the $NN_W$ using labels generated by the teacher without considering their confidence score, i.e. $FWL \setminus \Sigma$. This means we just augmented the fine-tuning process by generating a fine-tuning set using teacher which is better than $\mathcal{D}_s$ in terms of quantity and $\mathcal{D}_w$ in terms of quality. This baseline is equivalent to setting $\beta = 0$ in Equation 1. However, we see a big jump in performance when we use FWL to include the estimated label quality from the teacher, leading to the best overall results.

### 3.3 SENTIMENT CLASSIFICATION

In sentiment classification, the goal is to predict the sentiment (e.g., positive, negative, or neutral) of a sentence. Each training sample $x$ consists of a sentence $s$ and its sentiment label $\tilde{y}$.

Table 2: Performance of the proposed FWL approach and baseline methods for sentiment classification task. $\blacktriangle^i$ indicates that the improvements with respect to the baseline#$i$ are statistically significant, at the 0.05 level using the paired two-tailed t-test, with Bonferroni correction.

| | Method | SemEval-14 | SemEval-15 |
|---|---|---|---|
| 1 | $WA_{Lexicon}$ | 0.5141 | 0.4471 |
| 2 | $NN_W$ | $0.6719^{\blacktriangle 137}$ | $0.5606^{\blacktriangle 1}$ |
| 3 | $NN_S$ | $0.6307^{\blacktriangle 1}$ | $0.5811^{\blacktriangle 12}$ |
| 4 | $NN_{S+/W}$ | $0.7032^{\blacktriangle 1237}$ | $0.6319^{\blacktriangle 1237}$ |
| 5 | $NN_{W \rightarrow S}$ | $0.7080^{\blacktriangle 1237}$ | $0.6441^{\blacktriangle 1237}$ |
| 6 | $NN_{W^\omega \rightarrow S}$ | $0.7166^{\blacktriangle 12347}$ | $0.6603^{\blacktriangle 123457}$ |
| 7 | $FWL_{unsuprep}$ | $0.6588^{\blacktriangle 13}$ | $0.6954^{\blacktriangle 123}$ |
| 8 | $FWL \setminus \Sigma$ | $0.7202^{\blacktriangle 123457}$ | $0.6590^{\blacktriangle 123457}$ |
| 9 | FWL | $\mathbf{0.7470}^{\blacktriangle 12345678}$ | $\mathbf{0.6830}^{\blacktriangle 12345678}$ |
| 10 | $SemEval^{Best}$ | 0.7162 (Rouvier & Favre, 2016) | 0.6618 (Deriu et al., 2016) |

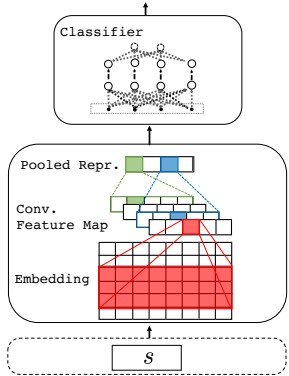

Figure 4: The student for the sentiment classification task.

**The student** for the sentiment classification task is a convolutional model which has been shown to perform best on the dataset we used (Deriu et al., 2017; Severyn & Moschitti, 2015a;b; Deriu et al., 2016). The first layer of the network learns the function $\psi(.)$ which maps input sentence $s$ to a dense vector as its representation. The inputs are first passed through an embedding layer mapping the sentence to a matrix $S \in \mathbb{R}^{m \times |s|}$, followed by a series of 1d convolutional layers with max-pooling. The representation layer is followed by feed-forward layers and a softmax output layer which returns the probability distribution over all three classes. Figure 4 presents the general schema of the architecture of the student. See Appendix B.2 for more details.

**The teacher** for this task is modeled by a $\mathcal{GP}$. See Appendix C for more details.

**The weak annotator** is a simple unsupervised lexicon-based method (Hamdan et al., 2013; Kiritchenko et al., 2014), which estimate a distribution over sentiments for each sentence, based on sentiment labels of its terms. More details are provided in Appendix D.2.

Specification of the data with weak labels and data with true labels along with the detailed experimental setup are given in Appendix E.3.

**Results and Discussion** We report Macro-F1, the official SemEval metric, in Table 2. We see that the proposed FWL is the best performing approach.

For this task, since the amount of data with true labels are larger compared to the ranking task, the performance of $NN_S$ is acceptable. Alternately sampling from weak and strong data gives better results. Pretraining on weak labels then fine-tuning the network on true labels, further improves the performance. Weighting the gradient updates from weak labels during pretraining and fine-tuning the network with true labels, i.e. $NN_{W^\omega \rightarrow S}$ seems to work quite well in this task. For this task, like ranking task, learning the representation in an unsupervised task independent fashion, i.e. $FWL_{unsuprep}$, does not lead to good results compared to the FWL. Similar to the ranking task, fine-tuning $NN_S$ based on labels generated by $\mathcal{GP}$ instead of data with true labels, regardless of the confidence score, works better than standard fine-tuning.

Besides the baselines, we also report the best performing systems which are also convolution-based models (Rouvier & Favre 2016 on SemEval-14; Deriu et al. 2016 on SemEval-15). Using FWL and taking the confidence into consideration outperforms the best systems and leads to the highest reported results on both datasets.

## 4 ANALYSIS

In this section, we provide further analysis of FWL by investigating the bias-variance trade-off and the learning rate.

### 4.1 HANDLING THE BIAS-VARIANCE TRADE-OFF

As mentioned in Section 2, $\beta$ is a hyperparameter that controls the contribution of weak and strong data to the training procedure. In order to investigate its influence, we fixed everything in the model and ran the fine-tuning step with different values of $\beta \in \{0.0, 0.1, 1.0, 2.0, 5.0\}$ in all the experiments.

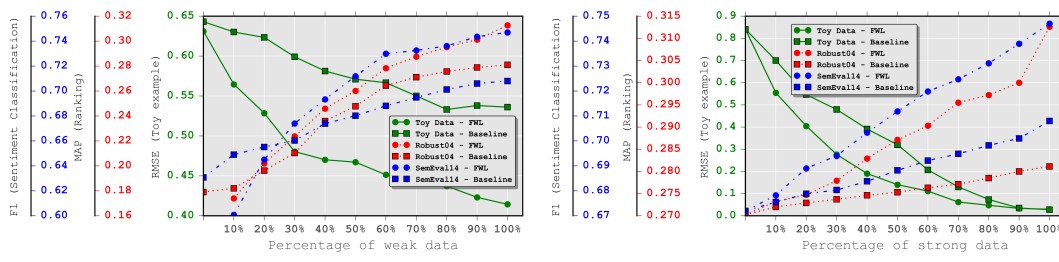

(a) Models trained on different amount weak data.   (b) Models trained on different amount of strong data.

Figure 6: Performance of FWL and the baseline model trained on different amount of data.

Figure 5 illustrates the performance on the ranking (on Robust04 dataset) and sentiment classification tasks (on SemEval14 dataset). For both sentiment classification and ranking, $\beta = 1$ gives the best results (higher scores are better). We also experimented on the toy problem with different values of $\beta$ in three cases: 1) having 10 observations from the true function (same setup as Section 3.1), marked as "Toy Data" in the plot, 2) having only 5 observations from the true function, marked as "Toy Data *" in the plot, and 3)

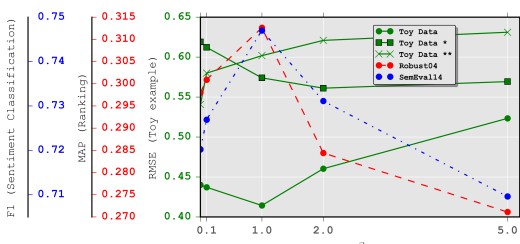

Figure 5: Effect of different values for $\beta$.

having $f(x) = x + 1$ as the weak function, which is an extremely bad approximator of the true function, marked as "Toy Data **" in the plot. For the "Toy Data" experiment, $\beta = 1$ turned out to be optimal (here, lower scores are better). However, for "Toy Data *", where we have an extremely small number of observations from the true function, setting $\beta$ to a higher value acts as a regularizer by relying more on weak signals, and eventually leads to better generalization. On the other hand, for "Toy Data **", where the quality of the weak annotator is extremely low, lower values of $\beta$ put more focus on the true observations. Therefore, $\beta$ lets us control the bias-variance trade-off in these extreme cases.

## 4.2  A GOOD TEACHER IS BETTER THAN MANY OBSERVATIONS

We now look at the rate of learning for the student as the amount of training data is varied. We performed two types of experiments for all tasks: In the first experiment, we use all the available strong data but consider different percentages of the entire weak dataset. In the second experiment, we fix the amount of weak data and provide the model with varying amounts of strong data. We use standard fine-tuning with similar setups as for the baseline models. Details on the experiments for the toy problem are provided in Appendix E.1.

Figure 6 presents the results of these experiments. In general, for all tasks and both setups, the student learns faster when there is a teacher. One caveat is in the case where we have a very small amount of weak data. In this case the student cannot learn a suitable representation in the first step, and hence the performance of FWL is pretty low, as expected. It is highly unlikely that this situation occurs in reality as obtaining weakly labeled data is much easier than strong data.

The empirical observation of Figure 6 that our model learns more with less data can also be seen as evidence in support of another perspective to FWL, called *learning using privileged information* (Vapnik & Izmailov, 2015). We elaborate more on this connection in Appendix F.

## 4.3  SENSITIVITY OF THE FWL TO THE QUALITY OF THE WEAK ANNOTATOR

Our proposed setup in FWL requires defining a so-called "weak annotator" to provide a source of weak supervision for unlabelled data. In Section 4.1 we discussed the role of parameter $\beta$ for controlling the bias-variance trade-off by trying two weak annotators for the toy problem. Now, in this section, we study how the quality of the weak annotator may affect the performance of the FWL, for the task of document ranking as a real-world problem.

To do so, besides BM25 (Robertson & Zaragoza, 2009), we use three other weak annotators:

vector space model (Salton & Yang, 1973) with binary term occurrence (BTO) weighting schema and vector space model with TF-IDF weighting schema, which are both weaker than BM25, and BM25+RM3 (Abdul-jaleel et al., 2004) that uses RM3 as the pseudo-relevance feedback method on top of BM25, leading to better labels.

Figure 7 illustrates the performance of these four weak annotators in terms of their mean average precision (MAP) on the test data, versus the performance of FWL given the corresponding weak annotator. As it is expected, the performance of FWL depends on the quality of the employed weak annotator. The percentage of improvement of FWL over its corresponding weak annotator on the test data is also presented in Figure 7. As can be seen, the better the performance of the weak annotator is, the less the improvement of the FWL would be.

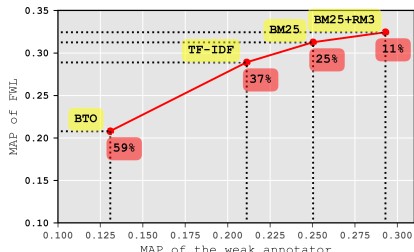

Figure 7: Performance of FWL versus performance of the corespondence weak annotator in the document ranking task, on Robust04 dataset.

## 4.4 FROM MODIFYING THE LEARNING RATE TO WEIGHTED SAMPLING

FWL provides confidence score based on the certainty associated with each generated label $\bar{y}_t$, given sample $x_t \in \mathcal{D}_{sw}$. We can translate the confidence score as how likely including $(x_t, \bar{y}_t)$ in the training set for the student model improves the performance, and rather than using this score as the multiplicative factor in the learning rate, we can use it to bias sampling procedure of mini-batches so that the frequency of training samples are proportional to the confidence score of their labels.

We design an experiment to try FWL with this setup (FWL$_s$), in which we keep the architectures of the student and the teacher and the procedure of the first two steps of the FWL fixed, but we changed the step 3 as follows: Given the soft dataset $\mathcal{D}_{sw}$, consisting of $x_t$, its label $\bar{y}_t$ and the associated confidence score generated by the teacher,

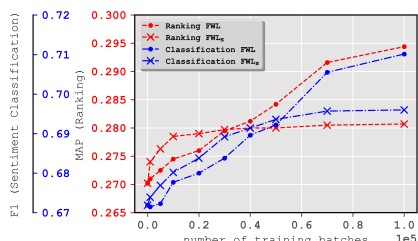

Figure 8: Performance of FWL and FWL$_s$ with respect to different batch of data for the task of document ranking (Robust04 dataset) and sentiment classification (SemEval14 dataset).

we normalize the confidence scores over all training samples and set the normalized score of each sample as its probability to be sampled. Afterward, we train the student model by mini-batches sampled from this set with respect to the probabilities associated with each sample, but without considering the original confidence scores in parameter updating. This means the more confident the teacher is about the generated label for each sample, the more chance that sample has to be seen by the student model.

Figure 8 illustrates the performance of both FWL and FWL$_s$ trained on different amount of data sampled from $\mathcal{D}_{sw}$, in the document ranking and sentiment classification tasks. As can be seen, compared to FWL, the performance of FWL$_s$ increases rapidly in the beginning but it slows down afterward. We have looked into the sampling procedure and noticed that the confidence scores provided by the teacher form a rather skewed distribution and there is a strong bias in FWL$_s$ toward sampling from data points that are either in or closed to the points in $\mathcal{D}_s$, as $\mathcal{GP}$ has less uncertainty around these points and the confidence scores are high. We observed that the performance of FWL$_s$ gets closer to the performance of FWL after many epochs, while FWL had already a log convergence. The skewness of the confidence distribution makes FWL$_s$ to have a tendency for more exploitation than exploration, however, FWL has more chance to explore the input space, while it controls the effect of updates on the parameters for samples based on their merit.

## 5 RELATED WORK

In this section, we position our FWL approach relative to related work.

Learning from imperfect labels has been thoroughly studied in the literature (Frénay & Verleysen, 2014). The imperfect (weak) signal can come from non-expert crowd workers, be the output of other models that are weaker (for instance with low accuracy or coverage), biased, or models trained on data from different related domains. Among these forms, in the distant supervision setup, a heuristic labeling rule (Deriu et al., 2016; Severyn & Moschitti, 2015b) or function (Dehghani et al., 2017d)

which can be relying on a knowledge base (Mintz et al., 2009; Min et al., 2013; Han & Sun, 2016) is employed to devise noisy labels.

Learning from weak data sometimes aims at encoding various forms of domain expertise or cheaper supervision from lay annotators. For instance, in the structured learning, the label space is pretty complex and obtaining a training set with strong labels is extremely expensive, hence this class of problems leads to a wide range of works on learning from weak labels (Roth, 2017). Indirect supervision is considered as a form of learning from weak labels that is employed in particular in the structured learning, in which a companion binary task is defined for which obtaining training data is easier (Chang et al., 2010; Raghunathan et al., 2016). In the response-based supervision, the model receives feedback from interacting with an environment in a task, and converts this feedback into a supervision signal to update its parameters (Roth, 2017; Clarke et al., 2010; Riezler et al., 2014). Constraint-based supervision is another form of weak supervision in which constraints that are represented as weak label distributions are taken as signals for updating the model parameters. For instance, physics-based constraints on the output (Stewart & Ermon, 2017) or output constraints on execution of logical forms (Clarke et al., 2010).

In the proposed FWL model, we can employ these approaches as the weak annotator to provide imperfect labels for the unlabeled data, however, a small amount of data with strong labels is also needed, which put our model in the class of semi-supervised models. In the semi-supervised setup, some ideas were developed to utilize weakly or even unlabeled data. For instance, the idea of self(incremental)-training (Rosenberg et al., 2005), pseudo-labeling (Lee, 2013; Hinton et al., 2014), and Co-training (Blum & Mitchell, 1998) are introduced for augmenting the training set by unlabeled data with predicted labels. Some research used the idea of self-supervised (or unsupervised) feature learning (Noroozi & Favaro, 2016; Dosovitskiy et al., 2016; Donahue et al., 2017) to exploit different labelings that are freely available besides or within the data, and to use them as intrinsic signals to learn general-purpose features. These features, that are learned using a proxy task, are then used in a supervised task like object classification/detection or description matching.

As a common approach in semi-supervised learning, the unlabeled set can be used for learning the distribution of the data. In particular for neural networks, greedy layer-wise pre-training of weights using unlabeled data is followed by supervised fine-tuning (Hinton et al., 2006; Deriu et al., 2017; Severyn & Moschitti, 2015b;a; Go et al., 2009). Other methods learn unsupervised encoding at multiple levels of the architecture jointly with a supervised signal (Ororbia II et al., 2015; Weston et al., 2012).

Alternatively, some noise cleansing methods have been proposed to remove or correct mislabeled samples (Brodley & Friedl, 1999). There are some studies showing that weak or noisy labels can be leveraged by modifying the loss function (Reed et al., 2015; Patrini et al., 2017; 2016; Vahdat, 2017) or changing the update rule to avoid imperfections of the noisy data (Malach & Shalev-Shwartz, 2017; Dehghani et al., 2017b;c).

One direction of research focuses on modeling the pattern of the noise or weakness in the labels. For instance, methods that use a generative model to correct weak labels such that a discriminative model can be trained more effectively (Ratner et al., 2016; Rekatsinas et al., 2017; Varma et al., 2017). Furthermore, methods that aim at capturing the pattern of the noise by inserting an extra layer (Goldberger & Ben-Reuven, 2017) or a separate module tries to infer better labels from noisy ones and use them to supervise the training of the network (Sukhbaatar et al., 2015; Veit et al., 2017; Dehghani et al., 2017b). Our proposed FWL can be categorized in this class as the teacher tries to infer better labels and provide certainty information which is incorporated as the update rule for the student model.

## 6  CONCLUSION

Training neural networks using large amounts of weakly annotated data is an attractive approach in scenarios where an adequate amount of data with true labels is not available, a situation which often arises in practice. In this paper, we introduced fidelity-weighted learning (FWL), a new student-teacher framework for semi-supervised learning in the presence of weakly labeled data. We applied FWL to document ranking and sentiment classification, and empirically verified that FWL speeds up the training process and improves over state-of-the-art semi-supervised alternatives.

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

APPENDICES

We moved additional details to the appendices in order to keep the main text focused on the overall idea of the Fidelity-Weighted Learning approach. Specifically, we include further details on the clustered Gaussian process approach (Appendix A); on the student network architectures (Appendix B); on the teacher Gaussian process model (Appendix C); on the weak annotators (Appendix D); on the experimental data and setup (Appendix E); and on the connection to "learning with privileged information" (Appendix F).

## A    DETAILED DESCRIPTION OF CLUSTERED GP

We suggest using several $\mathcal{GP} = \{GP_{c_i}\}$ to explore the entire data space more effectively. Even though inducing points and stochastic methods make $\mathcal{GP}$s more scalable we still observed poor performance when the entire dataset was modeled by a single $\mathcal{GP}$. Therefore, the reason for using multiple $\mathcal{GP}$s is mainly empirical inspired by (Shen et al., 2006) which is explained in the following:

We used Sparse Gaussian Process implemented in GPflow. The algorithm is scalable in the sense that it is not $O(N^3)$ as original $\mathcal{GP}$ is. It introduces inducing points in the data space and defines a variational lower bound for the marginal likelihood. The variational bound can now be optimized by stochastic methods which make the algorithm applicable in large datasets. However, the tightness of the bound depends on the location of inducing points which are found through the optimization process. We empirically observed that a single $\mathcal{GP}$ does not give a satisfactory accuracy on left-out test dataset. We hypothesized that this can be due to the inability of the algorithm to find good inducing points when the number of inducing points is restricted to just a few. Then we increased the number of inducing points $M$ which trades off the scalability of the algorithm because it scales with $O(NM^2)$. Moreover, apart from scalability which is partly solved by stochastic methods, we argue that the structure of the entire space may not be explored well by a single $\mathcal{GP}$ and its inducing points. We guess this can be due to the observation that our datasets are distributed in a highly sparse way within the high dimensional embedding space. We also tried to cure the problem by means of PCA to reduce input dimensions and give a denser representation, but it did not result in a considerable improvement. The results are presented in Tabel 3.

Table 3:  Performance of FWL using a single $\mathcal{GP}$, a single $\mathcal{GP}$ after applying PCA on the input data, and the clustered $\mathcal{GP}$ as the teacher.

| Method | Document Ranking | | | | Sentiment Classification | |
|---|---|---|---|---|---|---|
| | Robust04 | | ClueWeb | | Robust04 | ClueWeb |
| | MAP | nDCG@20 | MAP | nDCG@20 | F1 | F1 |
| $FWL_{\mathcal{GP}}$ | 0.2614 | 0.4192 | 0.1205 | 0.2121 | 0.6904 | 0.6173 |
| $FWL_{PCA \to \mathcal{GP}}$ | 0.2864 | 0.4411 | 0.1331 | 0.2388 | 0.7022 | 0.6340 |
| $FWL_{Clustered\ \mathcal{GP}}$ | **0.3124** | **0.4607** | **0.1472** | **0.2453** | **0.7470** | **0.6830** |

We may be able to argue that clustered $\mathcal{GP}$ makes better use of the data structure roughly close to the idea of KISS-GP (Wilson & Nickisch, 2015). In inducing point methods, it is normally assumed that $M \ll N$ ($M$ is the number of inducing points and $N$ is the number of training samples) for computational and storage saving. However, we have this intuition that few number of inducing points make the model unable to explore the inherent structure of data. By employing several GPs, we were able to use a large number of inducing points even when $M > N$ ($M$ is the total number of inducing points) which seemingly better exploits the structure of datasets. Because our work was not aimed to be a close investigation of GP, we considered clustered $\mathcal{GP}$ as the engineering side of the work which is a tool to give us a measure of confidence. Other tools such as a single $\mathcal{GP}$ with inducing points that form a Kronecker or Toeplitz covariance matrix are also conceivable. Therefore, we do not of course claim that we have proposed a new method of inference for GPs. Here is practical description of clustered $\mathcal{GP}$ algorithm:

*Clustered $\mathcal{GP}$*: Let $N$ be the size of the dataset on which we train the teacher. Assume we allocate $K$ teachers to the entire data space. Therefore, each $\mathcal{GP}$ sees a dataset of size $n = N/K$. Then we use a simple clustering method (e.g. k-means) to find centroids of $K$ clusters $C_1, C_2, ..., C_K$ where $C_i$ consists of samples $\{x_{i,1}, x_{i,2}, ..., x_{i,n}\}$. We take the centroid $c_i$ of cluster $C_i$ as the representative sample for all its content. Note that $c_i$ does not necessarily belong to $\{x_{i,1}, x_{i,2}, ..., x_{i,n}\}$. We assign each cluster a $\mathcal{GP}$ trained by samples belonging to that cluster. More precisely, cluster $C_i$ is assigned a $\mathcal{GP}$ whose data points are $\{x_{i,1}, x_{i,2}, ..., x_{i,n}\}$. Because there is no dependency among different clusters, we train them in parallel to speed-up the procedure more.

The pseudo-code of the clustered $\mathcal{GP}$ is presented in Algorithm 2. When the main issue is computational resources (when the number of inducing points for each $\mathcal{GP}$ is large), we can first choose the number $n$ which is the maximum size of the dataset on which our resources allow to train a $\mathcal{GP}$, then find the number of clusters $K = N/n$ accordingly. The rest of the algorithm remains unchanged.

---

**Algorithm 2** Clustered Gaussian processes.

---

1: Let $N$ be the sample size, $n$ the sample size of each cluster, $K$ the number of clusters, and $c_i$ the center of cluster $i$.
2: Run K-means with $K$ clusters over all samples with true labels $\mathcal{D}_s = \{x_i, y_i\}$.

$$\text{K-means}(x_i) \to c_1, c_2, ..., c_K$$

where $c_i$ represents the center of cluster $C_i$ containing samples $D_s^{c_i} = \{x_{i,1}, x_{i,2}, ... x_{i,n}\}$.
3: Assign each of $K$ clusters a Gaussian process and train them in parallel to approximate the label of each sample.

$$
\begin{aligned}
\mathcal{GP}_{c_i}(\boldsymbol{m}_{\text{post}}^{c_i}, K_{\text{post}}^{c_i}) &= \mathcal{GP}(\boldsymbol{m}_{\text{prior}}, K_{\text{prior}}) | D_s^{c_i} = \{(\psi(x_{s,c_i}), y_{s,c_i})\} \\
T_{c_i}(x_t) &= g(\boldsymbol{m}_{\text{post}}^{c_i}(x_t)) \\
\Sigma_{c_i}(x_t) &= h(K_{\text{post}}^{c_i}(x_t, x_t))
\end{aligned}
$$

where $\mathcal{GP}_{c_i}$ is trained on $\mathcal{D}_s^{c_i}$ containing samples belonging to the cluster $c_i$. Other elements are defined in Section 2
4: Use trained teacher $T_{c_i}(.)$ to evaluate the soft label and uncertainty for samples from $\mathcal{D}_{sw}$ to compute $\eta_2(x_t)$ required for step 3 of Algorithm 1. We use $T(.)$ as a wrapper for all teachers $\{T_{c_i}\}$.

---

# B   DETAILED ARCHITECTURE OF THE STUDENTS

## B.1   RANKING TASK

For the ranking task, the employed student is proposed in (Dehghani et al., 2017d). The first layer of the network models function $\psi$ that learns the representation of the input data samples, i.e. $(q, d^+, d^-)$, and consists of three components: (1) an embedding function $\varepsilon : \mathcal{V} \to \mathbb{R}^m$ (where $\mathcal{V}$ denotes the vocabulary set and $m$ is the number of embedding dimensions), (2) a weighting function $\omega : \mathcal{V} \to \mathbb{R}$, and (3) a compositionality function $\odot : (\mathbb{R}^m, \mathbb{R})^n \to \mathbb{R}^m$. More formally, the function $\psi$ is defined as:

$$
\begin{aligned}
\psi(q, d^+, d^-) = [&\odot_{i=1}^{|q|}(\varepsilon(t_i^q), \omega(t_i^q)) \,\| \\
&\odot_{i=1}^{|d^+|}(\varepsilon(t_i^{d^+}), \omega(t_i^{d^+})) \,\| \\
&\odot_{i=1}^{|d^-|}(\varepsilon(t_i^{d^-}), \omega(t_i^{d^-}))\,],
\end{aligned}
\tag{2}
$$

where $t_i^q$ and $t_i^d$ denote the $i^{th}$ term in query $q$ respectively document $d$. The embedding function $\varepsilon$ maps each term to a dense $m$- dimensional real value vector, which is learned during the training phase. The weighting function $\omega$ assigns a weight to each term in the vocabulary. It has been shown that $\omega$ simulates the effect of inverse document frequency (IDF), which is an important feature in information retrieval (Dehghani et al., 2017d).

The compositionality function $\odot$ projects a set of $n$ embedding-weighting pairs to an $m$- dimensional representation, independent from the value of $n$:

$$\bigodot_{i=1}^{n}(\varepsilon(t_i), \omega(t_i)) = \frac{\sum_{i=1}^n \exp(\omega(t_i)) \cdot \varepsilon(t_i)}{\sum_{j=1}^n \exp(\omega(t_j))}, \tag{3}$$

which is in fact the normalized weighted element-wise summation of the terms' embedding vectors. Again, it has been shown that having global term weighting function along with embedding function improves the performance of ranking as it simulates the effect of inverse document frequency (IDF). In our experiments, we initialize the embedding function $\varepsilon$ with word2vec embeddings (Mikolov et al., 2013) pre-trained on Google News and the weighting function $\omega$ with IDF.

The representation layer is followed by a simple fully connected feed-forward network with $l$ hidden layers followed by a softmax which receives the vector representation of the inputs processed by the representation learning layer and outputs a prediction $\tilde{y}$. Each hidden layer $z_k$ in this network computes $z_k = \alpha(W_k z_{k-1} + b_k)$, where $W_k$ and $b_k$ denote the weight matrix and the bias term corresponding to the $k^{th}$ hidden layer and $\alpha(.)$ is the non-linearity. These layers follow a sigmoid output. We employ the cross entropy loss:

$$\mathcal{L}_t = \sum_{i \in B} [-y_i \log(\hat{y}_i) - (1 - y_i)\log(1 - \hat{y}_i)], \tag{4}$$

where $B$ is a batch of data samples.

## B.2  SENTIMENT CLASSIFICATION TASK

The student for the sentiment classification task is a convolutional model which has been shown to perform best in the dataset we used (Deriu et al., 2017; Severyn & Moschitti, 2015a;b; Deriu et al., 2016). The first layer of the network learns the function $\psi$ which maps input sentence $s$ to a vector as its representation consists of an embedding function $\varepsilon : \mathcal{V} \to \mathbb{R}^m$, where $\mathcal{V}$ denotes the vocabulary set and $m$ is the number of embedding dimensions.

This function maps the sentence to a matrix $S \in \mathbb{R}^{m \times |s|}$, where each column represents the embedding of a word at the corresponding position in the sentence. Matrix $S$ is passed through a convolution layer. In this layer, a set of $f$ filters is applied to a sliding window of length $h$ over $S$ to generate a feature map matrix $C$. Each feature map $c_i$ for a given filter $F$ is generated by $c_i = \sum_{k,j} S[i:i+h]_{k,j} F_{k,j}$, where $S[i:i+h]$ denotes the concatenation of word vectors from position $i$ to $i+h$. The concatenation of all $c_i$ produces a feature vector $c \in \mathbb{R}^{|s|-h+1}$. The vectors $c$ are then aggregated over all $f$ filters into a feature map matrix $C \in \mathbb{R}^{f \times (|s|-h+1)}$.

We also add a bias vector $b \in R^f$ to the result of a convolution. Each convolutional layer is followed by a non-linear activation function (we use ReLU(Nair & Hinton, 2010)) which is applied element-wise. Afterward, the output is passed to the max pooling layer which operates on columns of the feature map matrix $C$ returning the largest value: $pool(c_i) : \mathbb{R}^{1 \times (|s|-h+1)} \to \mathbb{R}$ (see Figure 4). This architecture is similar to the state-of-the-art model for Twitter sentiment classification from Semeval 2015 and 2016 (Severyn & Moschitti, 2015b; Deriu et al., 2016).

We initialize the embedding matrix with word2vec embeddings (Mikolov et al., 2013) pretrained on a collection of 50M tweets.

The representation layer then is followed by a feed-forward layer similar to the ranking task (with different width and depth) but with softmax instead of sigmoid as the output layer which returns $\hat{y}_i$, the probability distribution over all three classes. We employ the cross entropy loss:

$$\mathcal{L}_t = \sum_{i \in B} \sum_{k \in K} -y_i^k \log(\hat{y}_i^k), \tag{5}$$

where $B$ is a batch of data samples, and $K$ is a set of classes.

## C  DETAILED ARCHITECTURE OF THE TEACHERS

We use Gaussian Process as the teacher in all the experiments. For each task, either regression or (multi-class) classification, in order to generate soft labels, we pass the mean of $\mathcal{GP}$ through the same function $g(.)$ that is applied on the output of the student network for that task, e.g. softmax, or sigmoid. For binary classification or one dimensional regression, $\Sigma(x_t)$ is scalar and $h(.)$ is identity. For multi-class classification or multi-dimensional regression tasks, $h(.)$ is an aggregation function that takes variance over several dimensions and outputs a single measure of variance. As a reasonable choice, the aggregating function $h(.)$ in our sentiment classification task (three classes) is *mean* of variances over dimensions.

In the teacher, linear combinations of different kernels are used for different tasks in our experiments.

**Toy Problem:**  We use standard Gaussian process regression[2] with this kernel:

$$k(x_i, x_j) = k_{\mathrm{RBF}}(x_i, x_j) + k_{\mathrm{White}}(x_i, x_j) \tag{6}$$

**Document Ranking:**  We use sparse variational GP regression[3] (Titsias, 2009) with this kernel:

$$k(x_i, x_j) = k_{\mathrm{Matern3/2}}(x_i, x_j) + k_{\mathrm{Linear}}(x_i, x_j) + k_{\mathrm{White}}(x_i, x_j) \tag{7}$$

**Sentiment Classification:**  We use sparse variational GP for multiclass classification[4] (Hensman et al., 2015) with the following kernel:

$$k(x_i, x_j) = k_{\mathrm{RBF}}(x_i, x_j) + k_{\mathrm{Linear}}(x_i, x_j) + k_{\mathrm{White}}(x_i, x_j) \tag{8}$$

---

[2] http://gpflow.readthedocs.io/en/latest/notebooks/regression.html
[3] http://gpflow.readthedocs.io/en/latest/notebooks/SGPR_notes.html
[4] http://gpflow.readthedocs.io/en/latest/notebooks/multiclass.html

where,

$$k_{\text{RBF}}(x_i,x_j) = \exp\left(\frac{\|x_i - x_j\|^2}{2l^2}\right)$$

$$k_{\text{Matern3/2}}(x_i,x_j) = \left(1 + \frac{\sqrt{3}\|x_i - x_j\|}{l}\right)\exp\left(-\frac{\sqrt{3}\|x_i - x_j\|}{l}\right)$$

$$k_{\text{Linear}}(x_i,x_j) = \sigma_0^2 + x_i.x_j$$

$$k_{\text{White}}(x_i,x_j) = constant\_value, \quad \forall x_1 = x_2 \text{ and } 0 \text{ otherwise}$$

We empirically found $l = 1$ satisfying value for the length scale of RBF and Matern3/2 kernels. We also set $\sigma_0 = 0$ to obtain a homogeneous linear kernel. The constant value of $K_{White}(.,.)$ determines the level of noise in the labels. This is different from the noise in weak labels. This term explains the fact that even in true labels there might be a trace of noise due to the inaccuracy of human labelers.

We set the number of clusters in the clustered $\mathcal{GP}$ algorithm for the ranking task to 50 and for the sentiment classification task to 30.

## D    WEAK ANNOTATORS

### D.1    DOCUMENT RANKING

The weak annotator in the document ranking task is BM25 (Robertson & Zaragoza, 2009), a well-known unsupervised retrieval method. This method heuristically scores a given pair of query-document based on the statistics of their matched terms. In the pairwise document ranking setup, $\tilde{y}_i$ for a given sample $x_j = (q, d^+, d^-)$ is the probability of document $d^+$ being ranked higher than $d^-$: $\tilde{y}_i = P_{q,d^+,d^-} = s_{q,d^+}/s_{q,d^+} + s_{q,d^-}$, where $s_{q,d}$ is the score obtained from the weak annotator.

### D.2    SENTIMENT CLASSIFICATION

The weak annotator for the sentiment classification task is a simple lexicon-based method (Hamdan et al., 2013; Kiritchenko et al., 2014). We use SentiWordNet03 (Baccianella et al., 2010) to assign probabilities (positive, negative and neutral) for each token in set $\mathcal{D}_w$. We use a bag-of-words model for the sentence-level probabilities (i.e. just averaging the distributions of the terms), yielding a noisy label $\tilde{y}_i \in \mathbb{R}^{|K|}$, where $|K| = 3$ is the number of classes. We found empirically that using soft labels from the weak annotator works better than assigning a single hard label.

## E    DATA COLLECTION, PARAMETERS AND SETUP

### E.1    TOY PROBLEM

**Weak/True Data**  In all the experiments with the toy problem, we have randomly sampled 100 data points from the weak function and 10 data points from the true function. We introduce a small amount of noise to the observation of the true function to model the noise in the human labeled data.

**Setup**  The neural network employed in the toy problem experiments is a simple feed-forward network with the depth of 3 layers and width of 128 neurons per layer. We have used $tanh$ as the nonlinearity for the intermediate layers and a linear output layer. As the optimizer, we used Adam (Kingma & Ba, 2015) and the initial learning rate has been set to $0.001$. For the teacher in the toy problem, we fit only one $\mathcal{GP}$ on all the data points (i.e. no clustering). Also during fine-tuning, we set $\beta = 1$.

**Setup of experiments in Section 4.2**  We fixed everything in the model and tried running the fine-tuning step with different values for $\beta \in \{0.0, 0.1, 1.0, 2.0, 5.0\}$ in all the experiments. For the experiments on toy problem in Section 4.2, the reported numbers are averaged over 10 trials. In the first experiment (i.e. Figure 6a), the size of sampled data data is: $|\mathcal{D}_s| = 50$ and $|\mathcal{D}_w| = 100$ (Fixed) and for the second one (i.e. Figure 6a): $|\mathcal{D}_w| = 100$ and $|\mathcal{D}_s| = 10$ (fixed).

### E.2    RANKING TASK

**Collections**  We use two standard TREC collections for the task of ad-hoc retrieval: The first collection (*Robust04*) consists of 500k news articles from different news agencies as a homogeneous collection. The second collection (*ClueWeb*) is ClueWeb09 Category B, a large-scale web collection with over 50 million English documents,

which is considered as a heterogeneous collection. Spam documents were filtered out using the Waterloo spam scorer[5] (Cormack et al., 2011) with the default threshold 70%.

**Data with true labels** We take query sets that contain human-labeled judgments: a set of 250 queries (TREC topics 301–450 and 601–700) for the Robust04 collection and a set of 200 queries (topics 1-200) for the experiments on the ClueWeb collection. For each query, we take all documents judged as relevant plus the same number of documents judged as non-relevant and form pairwise combinations among them.

**Data with weak labels** We create a query set $Q$ using the unique queries appearing in the AOL query logs (Pass et al., 2006). This query set contains web queries initiated by real users in the AOL search engine that were sampled from a three-month period from March 2006 to May 2006. We applied standard pre-processing Dehghani et al. (2017d;a) on the queries: We filtered out a large volume of navigational queries containing URL substrings ("http", "www.", ".com", ".net", ".org", ".edu"). We also removed all non-alphanumeric characters from the queries. For each dataset, we took queries that have at least ten hits in the target corpus using our weak annotator method. Applying all these steps, We collect 6.15 million queries to train on in Robust04 and 6.87 million queries for ClueWeb. To prepare the weakly labeled training set $\mathcal{D}_w$, we take the top 1,000 retrieved documents using BM25 for each query from training query set $Q$, which in total leads to $\sim |Q| \times 10^6$ training samples.

**Setup** For the evaluation of the whole model, we conducted a 3-fold cross-validation. However, for each dataset, we first tuned all the hyper-parameters of the student in the first step on the set with true labels using batched GP bandits with an expected improvement acquisition function (Desautels et al., 2014) and kept the optimal parameters of the student fixed for all the other experiments. The size and number of hidden layers for the student is selected from $\{64,128,256,512\}$. The initial learning rate and the dropout parameter were selected from $\{10^{-3},10^{-5}\}$ and $\{0.0,0.2,0.5\}$, respectively. We considered embedding sizes of $\{300,500\}$. The batch size in our experiments was set to 128. We use ReLU (Nair & Hinton, 2010) as a non-linear activation function $\alpha$ in student. We use the Adam optimizer (Kingma & Ba, 2015) for training, and *dropout* (Srivastava et al., 2014) as a regularization technique.

At inference time, for each query, we take the top 2,000 retrieved documents using BM25 as candidate documents and re-rank them using the trained models. We use the Indri[6] implementation of BM25 with default parameters (i.e., $k_1 = 1.2$, $b = 0.75$, and $k_3 = 1,000$).

### E.3 SENTIMENT CLASSIFICATION TASK

**Collections** We test our model on the twitter message-level sentiment classification of SemEval-15 Task 10B (Rosenthal et al., 2015). Datasets of SemEval-15 subsume the test sets from previous editions of SemEval, i.e. SemEval-13 and SemEval-14. Each tweet was preprocessed so that URLs and usernames are masked.

**Data with true labels** We use train (9,728 tweets) and development (1,654 tweets) data from SemEval-13 for training and SemEval-13-test (3,813 tweets) for validation. To make your results comparable to the official runs on SemEval we us SemEval-14 (1,853 tweets) and SemEval-15 (2,390 tweets) as test sets (Rosenthal et al., 2015; Nakov et al., 2016).

**Data with weak labels** We use a large corpus containing 50M tweets collected during two months for both, training the word embeddings and creating the weakly annotated set $\mathcal{D}_w$ using the lexicon-based method explained in Section 3.3.

**Setup** Similar to the document ranking task, we tuned hyper-parameters for the student in the first step with respect to the true labels of the validation set using batched GP bandits with an expected improvement acquisition function (Desautels et al., 2014) and kept the optimal parameters fixed for all the other experiments. The size and number of hidden layers for the classifier and is selected from $\{32,64,128\}$. We tested the model with both, 1 and 2 convolutional layers. The number of convolutional feature maps and the filter width is selected from $\{200,300\}$ and $\{3,4,5\}$, respectively. The initial learning rate and the dropout parameter were selected from $\{1E-3,1E-5\}$ and $\{0.0,0.2,0.5\}$, respectively. We considered embedding sizes of $\{100,200\}$ and the batch size in these experiments was set to 64. ReLU (Nair & Hinton, 2010) is used as a non-linear activation function in student. Adam optimizer (Kingma & Ba, 2015) is used for training, and *dropout* (Srivastava et al., 2014) as a regularizer.

## F CONNECTION WITH VAPNIK'S LEARNING USING PRIVILEGED INFORMATION

In this section, we highlight the connections of our work with Vapnik's *learning using privileged information* (LUPI) (Vapnik & Vashist, 2009; Vapnik & Izmailov, 2015). FWL makes use of information from a small set of correctly labeled data to improve the performance of a semi-supervised learning algorithm. The main idea behind LUPI comes from the fact that humans learn much faster than machines. This can be due to the role that an

---

[5] http://plg.uwaterloo.ca/~gvcormac/clueweb09spam/
[6] https://www.lemurproject.org/indri.php

*Intelligent Teacher* plays in human learning. In this framework, the training data is a collection of triplets

$$\{(x_1,y_1,x_1^*),...,(x_n,y_n,x_n^*)\}{\sim}P^n(x,y,x^*) \tag{9}$$

where each $(x_i,y_i)$ is a pair of feature-label and $x_i^*$ is the additional information provided by an intelligent teacher to ease the learning process for the student. Additional information for each $(x_i,y_i)$ is available only during training time and the learning machine must only rely on $x_i$ at test time. The theory of LUPI studies how to leverage such a teaching signal $x_i^*$ to outperform learning algorithms utilizing only the normal features $x_i$. For example, MRI brain images can be augmented with high-level medical or even psychological descriptions of Alzheimer's disease to build a classifier that predicts the probability of Alzheimer's disease from an MRI image at test time. It is known from statistical learning theory (Vapnik, 1998) that the following bound for test error is satisfied with probability $1-\delta$:

$$R(f){\leq}R_n(f){+}O\left(\left(\frac{|\mathcal{F}|_{VC}{-}\log\sigma}{n}\right)^{\alpha}\right), \tag{10}$$

where $R_n(f)$ denotes the training error over $n$ samples, $|\mathcal{F}|_{VC}$ is the VC dimension of the space of functions from which $f$ is chosen, and $\alpha \in [0.5,1]$. When the classes are not *separable*, $\alpha = 0.5$ i.e. the machine learns at a slow rate of $O(n^{-1/2})$. For easier problems where classes are *separable*, $\alpha = 1$ resulting in a learning rate of $O(n^{-1})$. The difference between these two cases is severe. The same error bound achieved for a separable problem with 10 thousand data points is only obtainable for a non-separable problem when 100 million data points are provided. This is prohibitive even when obtaining large datasets is not so costly. The theory of LUPI shows that an intelligent teacher can reduce $\alpha$ resulting in a faster learning process for the student. In this paper, we proposed a *teacher-student* framework for semi-supervised learning. Similar to LUPI, in FWL a student is supposed to solve the main prediction task while an intelligent teacher provides additional information to improve its learning. In addition, we first train the student network so that it obtains initial knowledge of weakly labeled data and learns a good data representation. Then the teacher is trained on truly labeled data enjoying the representation learned by the student. This extends LUPI in a way that the teacher provides privileged information that is most useful for the current state of student's knowledge. FWL also extends LUPI by introducing several teachers each of which is specialized to correct student's knowledge related to a specific region of the data space.

Figure 6(a) provides evidence for the assumption that privileged information in our task can accelerate the learning process of the student. It shows how the privileged information from an intelligent teacher affects the exponent $\alpha$ of the error bound in Equation 10. Figure 6(b) shows the test error for various number of samples $|\mathcal{D}_s|$ with true label. As expected, In both extremes where $|\mathcal{D}_s|$ is too small or too large, the performance of our model becomes close to the models without a teacher. The reason is that student has enough strong samples to learn a good model of true function. In more realistic cases where $|\mathcal{D}_s| \ll |\mathcal{D}_w|$ but $|\mathcal{D}_s|$ is still large enough to be informative about $|\mathcal{D}_w|$, our model gives a lower test error than models without the intelligent teacher.

The theory of LUPI was first developed and proved for support vector machines by Vapnik as a method for knowledge transfer. Hinton introduced *Dark knowledge* as a spiritually close idea in the context of neural networks (Hinton et al., 2006). He proposed to use a large network or an ensemble of networks for training and a smaller network at test time. It turned out that compressing knowledge of a large system into a smaller system can improve the generalization ability. It was shown in (Lopez-Paz et al., 2016) that dark knowledge and LUPI can be unified under a single umbrella, called *generalized distillation*. The core idea of these models is *machines-teaching-machines*. As the name suggests, a machine is learning the knowledge embedded in another machine. In our case, student is correcting his knowledge by receiving privileged information about label uncertainty from teacher.

Our framework extends the core idea of LUPI in the following directions:

- Trainable teacher: It is often assumed that the teacher in LUPI framework has some additional true information. We show that when this extra information is not available, one can still use the LUPI setup and define an implicit teacher whose knowledge is learned from the true data. In this approach, the performance of the final student-teacher system depends on a clever answer to the following question: which information should be considered as the privileged knowledge of teacher.
- Bayesian teacher: The proposed teacher is Bayesian. It provides posterior uncertainty of the label of each sample.
- Mutual representation: We introduced module $\psi(.)$ which learns a mutual embedding (representation) for both student and teacher. This is in particular interesting because it defines a two-way channel between teacher and student.
- Multiple teachers: We proposed a scalable method to introduce several teachers such that each teacher is specialized in a particular region of the data space.

