# OpenReview forum: "Fidelity-Weighted Learning"
_ICLR.cc/2018/Conference — Accept (Poster)_

### Official Review · AnonReviewer3 · 2017-11-27
**From modifying learning rates to weighting samples with an estimate of uncertainty in label annotations?**

**Rating:** 5
**Confidence:** 4

**Review:**

The problem of interest is to train deep neural network models with few labelled training samples. The specific assumption is there is a large pool of unlabelled data, and a heuristic function that can provide label annotations, possibly with varying levels of noises, to those unlabelled data. The adopted learning model is of a student/teacher framework as in privileged learning/knowledge distillation/model compression, and also machine teaching. The student (deep neural network) model will learn from both labelled and unlabelled training data with the labels provided by the teacher (Gaussian process) model. The teacher also supplies an uncertainty estimate to each predicted label. How about the heuristic function? This is used for learning initial feature representation of the student model. Crucially, the teacher model will also rely on these learned features. Labelled data and unlabelled data are therefore lie in the same dimensional space.

Specific questions to be addressed:
1)	Clustering of strongly-labelled data points. Thinking about the statement “each an expert on this specific region of data space”, if this is the case, I am expecting a clustering for both strongly-labelled data points and weakly-labelled data points. Each teacher model is trained on a portion of strongly-labelled data, and will only predict similar weakly-labelled data. On a related remark, the nice side-effect is not right as it was emphasized that data points with a high-quality label will be limited. As well, GP models, are quite scalable nowadays (experiments with millions to billions of data points are available in recent NIPS/ICML papers, though, they are all rely on low dimensionality of the feature space for optimizing the inducing point locations).  It will be informative to provide results with a single GP model.
2)	From modifying learning rates to weighting samples. Rather than using uncertainty in label annotation as a multiplicative factor in the learning rate, it is more “intuitive” to use it to modify the sampling procedure of mini-batches (akin to baseline #4); sample with higher probability data points with higher certainty. Here, experimental comparison with, for example, an SVM model that takes into account instance weighting will be informative, and a student model trained with logits (as in knowledge distillation/model compression).

---

> ### Author Response · Authors · 2017-12-25
> **Response to reviewer #3 - part 2**
>
>
> Regarding the point: “I am expecting a clustering for both strongly-labelled data points and weakly-labelled data points”: In FWL, during training, in step#2, we train multiple GPs as the teacher using only the data with “strong labels” which is a rather small set. In step#3, we go through both data with strong and weak labels and for each data point, we assign each point to a teacher based on the centroid of the teacher’s corresponding cluster. Therefore, each teacher predicts the new label in its territory. The predicted labels are almost the same as the original labels for the strongly labeled points and hopefully better labels for the weakly labeled data points. The confidence for the newly labeled point is also reported by its corresponding GP (teacher).
> Clustering the weakly-labeled data points means having multiple student models as well. However,  during the recall, we do not want to have multiple students which is computationally and space-wise prohibitive. Having a separate student corresponding to each teacher prevent the makes each student almost blind with respect to other clusters which is not desirable. The single student defined in our framework enables it to have a holistic view of the entire input space. We want our main task to be solved by a single student which is assumed expressive enough. The entire framework is designed to help this single student to settle on a better local optimum enjoying multiple teachers in the distillation framework. One important point here is that in FWL the teacher can be implemented by any predictor that can provide uncertainty for its prediction [1]. Even though, we resort to GP and a small strongly labeled dataset to capture this uncertainty, we should argue that the concept is applicable also when the uncertainty signal is provided from outside the dataset.
> -----
> [1] Anonymous, Deep Neural Networks as Gaussian Processes, under submission at ICLR2018, https://openreview.net/forum?id=B1EA-M-0Z
>
> Q2- From modifying learning rates to weighting samples. Rather than using uncertainty in label annotation as a multiplicative factor in the learning rate, it is more “intuitive” to use it to modify the sampling procedure of mini-batches (akin to baseline #4); sample with higher probability data points with higher certainty. Here, experimental comparison with, for example, an SVM model that takes into account instance weighting will be informative, and a student model trained with logits (as in knowledge distillation/model compression).
>
> A2- We think the suggested comparison is the case mainly when samples are seen by the model with the frequency proportional to the certainty of their label.
> We designed a new experiment in which, we kept the architectures of the student and the teacher and the procedure of the first two steps of the FWL fixed. We changed the step#3 as follows: For each sample in D_{sw} (dataset consisting of strongly and weakly labeled data points which are relabeled by the teacher and each label is associated with a confidence), we normalize the confidence scores for all training samples and set the normalized score of each sample as its probability to be sampled.  Afterwards, we train student model by sampling mini-batches from D_{sw} with respect to the probabilities associated with each sample, without considering their confidence as a multiplicative factor for the learning rate.
> This means that more confident the teacher is about the generated label for each sample, the more chance that sample has to be seen by the student model.
> We have added a new subsection, “4.4. From modifying the learning rates to weighted sampling”, to the revised manuscript to report our observations. Based on the results, compared to the original FWL, the performance of FWL with sampling increases rapidly in the beginning but it slows down afterward.  We have looked into the sampling procedure and noticed that the confidence scores provided by the teacher form a rather skewed distribution and there is a strong bias toward sampling from data points that are either in or closed to the points in the dataset with strong labels, as GP has less uncertainty around these points and the confidence scores are high. We observed that the performance of the FWL with sampling gets closer to the performance of FWL after many epochs, while FWL had already a long convergence. The skewness of the confidence distribution makes FWL with sampling to have a tendency for more exploitation than exploration, however, FWL has more chance to explore the input space, while it controls the effect of updates on the parameters for samples based on their merit.
> We believe FWL with sampling can be improved by having a better strategy for sampling from a skewed distribution or using approaches for active learning and selective sampling which is out of the scope of this paper.

---

> ### Author Response · Authors · 2017-12-25
> **Response to reviewer #3 - part 1**
>
> First of all, we would like to thank the reviewer for the valuable suggestions and comments. Here we respond to the questions one by one:
>
> Q1- Clustering of strongly-labelled data points. Thinking about the statement “each an expert on this specific region of data space”, if this is the case, I am expecting a clustering for both strongly-labelled data points and weakly-labelled data points. Each teacher model is trained on a portion of strongly-labelled data, and will only predict similar weakly-labelled data. On a related remark, the nice side-effect is not right as it was emphasized that data points with a high-quality label will be limited. As well, GP models, are quite scalable nowadays (experiments with millions to billions of data points are available in recent NIPS/ICML papers, though, they are all rely on low dimensionality of the feature space for optimizing the inducing point locations).  It will be informative to provide results with a single GP model.
>
> A1- Regarding clustered GP: We used Sparse Gaussian Process implemented in GPflow to build our entire implementation in tensorflow. As the respected reviewer has mentioned, the algorithm is scalable in the sense that it is not O(N^3) as original GP is. It introduces inducing points in the data space and defines a variational lower bound for the marginal likelihood. The variational bound can now be optimized by stochastic methods which make the algorithm applicable to large datasets. However, the tightness of the bound depends on the locations which are found through the optimization process. We empirically observed that a single GP does not give a satisfactory accuracy on left-out test dataset. We hypothesized that this can be due to the inability of the algorithm to find good inducing points when only a few of them are available. Then we increased the number of inducing points which trades off the scalability of the algorithm because it scales with O(NM^2) where M is the number of inducing points. We guess this can be due to the observation that our datasets are distributed in a highly sparse way within the high dimensional embedding space. We also tried to cure the problem by means of PCA to reduce input dimension but it did not result in a considerable improvement. Due to this empirical evidence, we clustered the truly labeled dataset and used a separate GP for each cluster. The overall performance of the algorithm improved and we may be able to argue that clustered GP makes use of the data structure roughly close to the idea of KISS-GP[1]. In inducing-point methods (with m inducing points and n training samples), normally it is assumed that m<<n for computational and storage saving. However, we have this intuition that few number of inducing points make the model unable to explore the inherent structure of data. By employing several GPs, we were able to use a large number of total inducing points even m>n which seemingly better exploits the structure of datasets. Because our work was not aimed to be a close investigation of GP, we considered clustered GP as the engineering side of the work which is a tool to give us a measure of confidence. Other tools such as a single GP with inducing points that form a Kronecker or Toeplitz covariance matrix are also conceivable. Therefore, we do not of course claim that we have proposed a new method of inference for GP because of the lack of theoretical reasoning for the use of multiple GPs. In the end, as was asked by the reviewer, the result for single GP is included  as a part of clustered GP section in the Appendix A (Detailed description of clustered GP) in the revised manuscript showing the effectiveness of the the initial clustering and local GPs. Moreover, an abstract of this response is also added to the in Appendix A and also main text to enlighten the reason for using clustered GP.
> -----
> [1] Wilson, A. and Nickisch, H., 2015, June. Kernel interpolation for scalable structured Gaussian processes (KISS-GP). In International Conference on Machine Learning (pp. 1775-1784).

---

### Official Review · AnonReviewer2 · 2017-11-28
**well written, easy to follow, and have good experimental study but lacks enough motivation and justification for the proposed method**

**Rating:** 6
**Confidence:** 3

**Review:**

The authors propose an approach for training deep learning models for situation where there is not enough reliable annotated data.  This algorithm can be useful because correct annotation of enough cases to train a deep model in many domains is not affordable.  The authors propose to combine a huge number of weakly annotated data with a small set of strongly annotated cases to train a model in a student-teacher framework. The authors evaluate their proposed methods on one toy problem and two real-world problems. The paper is well written, easy to follow, and have good experimental study.  My main problem with the paper is the lack of enough motivation and justification for the proposed method; the methodology seems pretty ad-hoc to me and there is a need for more experimental study to show how the methodology work. Here are some questions that comes to my mind:  (1) Why first building a student model only using the weak data and why not all the data together to train the student model? To me, it seems that the algorithm first tries to learn a good representation for which lots of data is needed and the weak training data can be useful but why not combing with the strong data? (2) What are the sensitivity of the procedure to how weakly the weak data are annotated (this could be studied using both toy example and real-world examples)? (3) The authors explicitly suggest using an unsupervised method (check Baseline no.1) to annotate data weakly? Why not learning the representation using an unsupervised learning method (unsupervised pre training)? This should be at least one of the baselines.
(4) the idea of using surrogate labels to learn representation is also not new. One example work is "Discriminative Unsupervised Feature Learning with Exemplar Convolutional Neural Networks". The authors didn't compare their method with this one.

---

> ### Author Response · Authors · 2017-12-25
> **Response to reviewer #2 - part 3**
>
>
> Q4- the idea of using surrogate labels to learn representation is also not new. One example work is "Discriminative Unsupervised Feature Learning with Exemplar Convolutional Neural Networks". The authors didn't compare their method with this one.
>
> A4- Thanks for pointing out this paper.  The referred work is based on self-supervised feature learning in which the idea is to exploit different labelings that are freely available besides or within the data by defining a surrogate task which uses the intrinsic signals to learn better (e.g. generic, robust, descriptive and invariant) features. The learned features are then transferred to be used for a supervised task (e.g. object classification or description matching). We argue that in FWL we do not learn representation through a proxy task. We learn the representation (and pretrain the student model) downstream of the main task, but with pseudo-labels (noisy labels).  Nonetheless, we can say that representation learning in step#1 is solving a surrogate task of approximating the expert knowledge, for which a noisy supervision signal is provided by the weak annotator.
> In the response to the previous question, we have added a baseline for the sentiment classification task in which a surrogate task is used to learn the representation (see A3).  Furthermore, we discussed the advantage of the current setup of the step#1, i.e. learning the representation downstream of a task same as the final target task that we want to solve but with lower accuracy in the labels. Here we again summarize them and add one more point:
> 1.	Using the main task with weak labels in step#1 leads to a representation that complies better with the target task.
> 2.	In the current setup of the step#1, we also pretrain the student model, so representation learning is actually part of student pre-training in a weakly supervised manner (in A3, we explained why this is needed).
> 3.	In addition to the above points, the definition of the surrogate task in self-supervision depends on the problem to be solved. For instance, if the surrogate task is defined such that it yields features invariant to color, it cannot be used to differentiate objects with different colors. However, in our setup the step#1 is seen as a surrogate task is inherently in accordance with the main task (in fact they are the same, but with different accuracy in the label space) and we do not need to think about the suitable surrogate task for the feature learning phase.
> Looking to the FWL from the perspective of self-supervised feature learning is pretty interesting and valuable to mention. We have added this point (Section 2, where we are explaining step#1) and the related papers (in related work section) to the revised version of the submission to include this point of view as well.

---

> ### Author Response · Authors · 2017-12-25
> **Response to reviewer #2 - part 2**
>
>
> Q3- The authors explicitly suggest using an unsupervised method (check Baseline no.1) to annotate data weakly? Why not learning the representation using an unsupervised learning method (unsupervised pre-training)? This should be at least one of the baselines.
>
> A3- The suggestion in this comment is to learn the representation of the data, in the first step, in an unsupervised manner.
> We have already tried this idea for both tasks, i.e. removing the first step and replacing it with learning the representation in an unsupervised (or  self-supervised feature learning) way: In the document ranking task, as the representation of documents and queries, we use weighted averaging over pre-trained embeddings of their words based on their inverse document frequency [1]. In the sentiment analysis task, we use skip-thoughts [2] which tries to estimate representation for sentences by defining a surrogate task in which given a sentence, the goal is to predict the sentence before and after, using autoencoders. We have used these representations as the input for the GP in step#2 and #3.  In both tasks, the performance drops dramatically. There might be two main reasons for that:
> 1.	Learning the representation of the input data downstream of the main task that we are going to solve leads to representations that are better suited to the FWL (in terms of the communications between teacher and student) compared to a task-independent unsupervised way.
> 2.	The other reason for losing performance by replacing the first step with learning representation in an unsupervised (self-supervised) way is that although the main goal of step#1 is to learn a representation of the data for the given task, we pretrain all the parameters of the student network (not just the representation layer) in that step. So as mentioned in the paper, in step#3 “[...] for data points where the teacher is not confident, we down-weight the training steps of the student. This means that at these points, we keep the student function as it was trained on the weak data in Step#1.” In summary, the first step initializes the layers of both embedding network and classification network.
> We have added the aforementioned experiments as extra baselines (baseline #7, FWL_unsuprep) to the paper and in the results and discussions (section 3.2 and 3.3), we elaborate more on the importance of the FWL setup for step#1.
> ------------
> [1] Mostafa Dehghani, Hamed Zamani, Aliaksei Severyn, Jaap Kamps, and W. Bruce Croft. Neural ranking models with weak supervision. In SIGIR’17, 2017.
> [2] Ryan Kiros, Yukun Zhu, Ruslan R Salakhutdinov, Richard Zemel, Raquel Urtasun, Antonio Torralba, and Sanja Fidler. Skip-thought vectors. NIP2015, 2015.

---

> ### Author Response · Authors · 2017-12-25
> **Response to reviewer #2 - part 1**
>
> First of all, we would like to thank the reviewer for the valuable suggestions and comments. Here we respond to the questions one by one:
>
> Q0- My main problem with the paper is the lack of enough motivation and justification for the A0- proposed method; the methodology seems pretty ad-hoc to me and there is a need for more experimental study to show how the methodology work.
>
> A0- As we have discussed in the introduction section of the manuscript, the motivation is making the efficient use of training data to achieve better performance on test data. The situation where training data consists of a small set with good labels and a large set with weak labels is fairly common. For instance, in large scale classification tasks (e.g. ImageNet), a large number of people annotate images via Amazon Mechanical Turk. We can assume the labels generated by AMT are weak because we are not sure whether the distant labelers were concentrated enough or in a good mood or not. In addition, we may have a smaller set of labelers who are experts and concentrated on the annotation task. In this sense, we can consider the labels generated by the second group as strong labels. Our proposed framework can be used in cases where this split of the dataset into {small and strong labels, large and weak labels} is possible.  We tested our framework on NLP and IR tasks because the weak labels can be generated by a well-known heuristic function. However, the weak labels can also be generated by a separate set of mass labelers whose performance is weaker than a small set of strong labelers as was pointed out in the answer to the previous question. As another example in the field of machine vision, we can think of a pre-trained weak classifier which acts based on hand crafted features like SIFT or HoG. This trained classifier can then be used to label a large set of images and assign each image a weak label. In this framework, SIFT-based classifier substitutes the heuristic weak annotator of our paper. The other components and the overall framework remain unchanged.
>
> Here are some questions that comes to my mind:
> Q1- Why first building a student model only using the weak data and why not all the data together to train the student model? To me, it seems that the algorithm first tries to learn a good representation for which lots of data is needed and the weak training data can be useful but why not combining with the strong data?
>
> A1- We had experiments when both weak and strong data is used to build the representation (let’s call it mixed setup).  For both tasks, we observed no statistically significant difference (based on paired two-tailed t-test) between the performance of mixed setup and the setup proposed in the manuscript where only weak data is used to build the representation. Here are the results of these experiments:
>
> Ranking task:
> [Robust04 dataset:  Map=0.3105   / nDCG@20=0.46211]
> [ClueWeb dataset:  Map=0.1456  / nDCG@20=0.2439]
>
> Sentiment Classification task:
> [SemEval-14: F1= 0.7474  ]
> [SemEval-15: F1= 0.6811 ]
>
> We think the overall scores do not change since the strong data will eventually contribute to the parameter updates of the representation learning layer of the student model in step#3 (in Figure 1.c, the representation layer in the student model benefits from the gradient updates of samples from D_{sw}, which includes data with strong labels as well). Considering the fact that the final scores do not change significantly, we choose the current exposition as it is more generic in the sense that student does not need to see the strong labels in the step#1. This is important especially when weak and strong data are not available together due to, for instance, privacy issues.
>
> Q2- What are the sensitivity of the procedure to how weakly the weak data are annotated (this could be studied using both toy example and real-world examples)?
>
> A2- In the original version of the submission, we have a small experiment in section “4.1 Handling The Bias-Variance Trade-off”,  in which instead of f(x) = 2sinc(x), we use f(x) = x + 1  as a weaker annotator and we observed worse performance in particular for high values of the parameter \beta. In this experiment,  we actually aimed at studying the effect of parameter \beta.  We agree that having analysis on the sensitivity of the FWL to the quality of the weak annotation is beneficial, so we added a subsection, “4.3. The sensitivity of the FWL to the Quality of the Weak Annotator”, to the revised version of the submission in which we discussed the performance of FWL on the task of ranking, given four weak annotators with different accuracies.  As it is expected, the performance of FWL depends on the quality of the employed weak annotator. We also observed that the better the performance of the weak annotator was, the less the improvement of FWL over its corresponding weak annotator on test data would be.

---

### Official Review · AnonReviewer1 · 2017-12-01
**Overall, a nice paper**

**Rating:** 7
**Confidence:** 4

**Review:**

This paper suggests a simple yet effective approach for learning with weak supervision. This learning scenario involves two datasets, one with clean data (i.e., labeled by the true function) and one with noisy data, collected using a weak source of supervision.  The suggested approach assumes a teacher and student networks, and builds the final representation incrementally, by taking into account the "fidelity" of the weak label when training the student at the final step. The fidelity score is given by the teacher, after being trained over the clean data, and it's used to build a cost-sensitive loss function for the students. The suggested method seems to work well on several document classification tasks.

Overall, I liked the paper.  I would like the authors to consider the following questions -

- Over the last 10 years or so, many different frameworks for learning with weak supervision were suggested (e.g., indirect supervision, distant supervision, response-based, constraint-based, to name a few).  First, I'd suggest acknowledging these works and discussing the differences to your work. Second - Is your approach applicable to these frameworks?  It would be an interesting to compare to one of those methods  (e.g., distant supervision for relation extraction using a knowledge base), and see if by incorporating fidelity score, results improve.

- Can this approach be applied to semi-supervised learning? Is there a reason to assume the fidelity scores computed by the teacher would not improve the student in a self-training framework?

- The paper emphasizes that the teacher uses the student's initial representation, when trained over the clean data.  Is it clear that this step in needed? Can you add an additional variant of your framework when the fidelity score are  computed by the teacher when trained from scratch? using different architecture than the student?

 - I went over the authors comments and I appreciate their efforts to help clarify the issues raised.

---

> ### Author Response · Authors · 2017-12-25
> **Response to reviewer #1 - part 2**
>
>
> Q5- The paper emphasizes that the teacher uses the student's initial representation when trained over the clean data.  Is it clear that this step in needed?  Can you add an additional variant of your framework when the fidelity sores are computed by the teacher when trained from scratch? using a different architecture than the student?
>
> A5- In the current model, first the representation of the data is learned by the student using weakly annotated data, then, using the learned representation, we fit the teacher on the data with strong (true) labels. Providing the teacher with the learned representation by the student has three main reasons:
> First of all, it has been shown that we can learn effective representation of the data if we have a large quantity of data available, this can be either by learning the distribution of the data using unlabeled example, or learning representation of the data downstream of the main task using a large set of weakly labeled data. However, for many tasks using just a small amount of data with true labels, we will not be able to model the underlying distribution of the data. Since in FWL, the teacher is trained only on data with strong labels (which is a small set), sharing the learned representation of the previous step alleviates this problem and the teacher can enjoy the learned knowledge from the large quantity of the weakly annotated data.
> In our setup, we make use of a Gaussian Process as the teacher. It is an interesting direction to search for meaningful kernels on structured data, i.e strings in our case. There exist some works to define such non-vectorial kernels that are designed by experts and are domain specific [2,3]. However, our goal here is to learn the representation along with solving the main classification task. Even though sme papers connect Gaussian process and deep neural networks, we are not aware of a reliable method for end to end training to learn the input features of GP better than the features learned by a neural network.  So we do not learn the representation of the data as part of the step#2, but borrow it from step#1. Likewise, we do not learn the kernels of the GPs and only learn the vectorial representation of their inputs.
> As another possible advantage of the current setup, we let the teacher see the data through the student's lens. This may in particular help the teacher, in step#3, to provide better annotation (and confidence) for the training of the student when the teacher is aware of the idea of the student about metric properties of the input space learned in step#1.  Note that the input representation of the student is trained in the step#3 and is not fully identical with that of the teacher which is kept fixed. We tested the case where the teacher used the input representation of the student in step#3 but the accuracy dropped considerably. We ascribe this observation to the covariate shift in the input of a trained GP.
>
> We made these points more explicit in the revised version of the submission (Section 2, where we are explaining step#2). A variant of FWL would be to use one (or more) neural network(s) as the teacher [1] to be able to learn representation in step#2, but we believe it is necessary for the teacher and student to both agree on the metric space they see in the input.
>
> [1] Anonymous, Deep Neural Networks as Gaussian Processes, under submission at ICLR2018, https://openreview.net/forum?id=B1EA-M-0Z
> [2] Eskin E, Weston J, Noble WS, Leslie CS. Mismatch string kernels for SVM protein classification. InAdvances in neural information processing systems 2003 (pp. 1441-1448).
> [3] Gärtner T. A survey of kernels for structured data. ACM SIGKDD Explorations Newsletter. 2003 Jul 1;5(1):49-58

---

> ### Author Response · Authors · 2017-12-25
> **Response to reviewer #1 - part 1**
>
> First of all, we would like to thank the reviewer for the valuable suggestions and comments. Here we respond to the questions one by one:
>
> Q1- Over the last 10 years or so, many different frameworks for learning with weak supervision were suggested.
> First, I'd suggest acknowledging these works and discussing the differences to your work.
>
> A1- In the revised manuscript, we’ve included some of the main works in the area of “learning with weak supervision” (some of which had been left out in the original submission due to the page limit) in the related work section and discussed how FWL is related to them.
>
> Q2- Second - Is your approach applicable to these frameworks?  It would be an interesting to compare to one of those methods (e.g., distant supervision for relation extraction using a knowledge base), and see if by incorporating fidelity score, results improve.
>
> A2- In general, in order to employ FWL, we need a large set of data with weak labels. These weak labels can be devised using methods like distant supervision, indirect supervision, constraint-based supervision, etc.
> In our paper, for instance, for the ranking task, we use a weak annotator based on a heuristic function that can be considered as a form of distant supervision, and for the classification task, as the weak annotation, we use labels in the word level to infer labels in the sentence level which can be kind of considered as an indirect supervision approach with a slightly different setup.
> The interesting question would be how different approaches for providing the weak annotation may affect the performance of FWL, and in a more general perspective, how sensitive is FWL to the quality of the weak annotations. In the original submission, in section “4.1 Handling The Bias-Variance Trade-off”, we included a simple analysis on how employing different weak annotators with different qualities (in terms of accuracy on test data) affects the performance of FWL in the toy problem. Since this point is also raised by one of the other reviewers, we add extra analysis to the revised version for the ranking task, “4.3. The sensitivity of the FWL to the Quality of the Weak Annotator”. The analysis shows that the achieved improvement by FWL over the weak annotator decreases in the presence of a more accurate weak annotator. The reason could be the hypothesis that a good annotator makes better use of data and leaves less room for improvement by the teacher.
>
> Q3- Can this approach be applied to semi-supervised learning?
>
> A3- Yes. In fact, the proposed approach is applicable in the semi-supervised setup, where we can define one or more so-called “weak annotators”, to provide additional (albeit noisy) sources of weak supervision for unlabeled data. This can be done based on heuristics rules, or using a “weaker” or biased classifiers trained on e.g. non-expert crowd-sourced data or data from different domains that are related, or distant supervision where an external knowledge source is employed to devise labels. Providing such weak annotations is possible for a large class of tasks that are considered to be solved in the semi-supervised learning setup.
>
> Q4- Is there a reason to assume the fidelity scores computed by the teacher would not improve the student in a self-training framework?
>
> A4- If we correctly understood, the question here is “In what circumstances does taking the confidence (fidelity) score by FWL into account yield no improvement or even hurt the performance of the student model, while it learns from weakly annotated data?”.
> This could be an interesting direction that merits more detailed investigation.  The failures of applying the confidence score are either estimating a high confidence score for a bad training label (case#1), or estimating a low confidence for a good training label (case#2). From the set of controlled experiments, we have done in particular on the toy problem, we found that probably due to the Bayesian nature of the teacher in FWL, case#1 is less likely to happen compared to case#2. The explanation is that generation of bad labels is mostly due to the lack of enough strong data to fit a good GP. When the number of data points on which the GP is fitted is extremely low, the uncertainty is high almost all over the space leading to low confidences. So, in most cases, bad labels come with fairly low confidence.
> In our design, case#2 would not happen for samples with strong labels (since GP is fitted on them and the uncertainty is almost zero at those points), however, the teacher might reject good weak examples by assigning a low confidence score to them. This is not a crucial situation as 1. generating extra weak examples is not expensive in our setup, 2. the rejected weak example has already contributed to the parameter updates of the student in the pretraining with its original weak label (step #1). Nonetheless, having lots of case#2 leads to slower convergence of the model during training.

---

### Author Response · Authors · 2018-01-04
**List of the main changes made in the revised version of the paper**

Here we summarise a list of the changes and additions we made to the revised version of our manuscript. Each part is explained in detail in the responses to each question of each reviewer under the corresponding comment.

- Section 2: providing more intuition and justification on the current setup of FWL in the description of the step#1 of the algorithm
- Section 2: quick pointer to the details of the clustered GP in the description of the step#2 of the algorithm
- Section 3: adding one more baseline and its corresponding results and related discussions
- Section 4: new small subsection: 4.3. Sensitivity of the FWL to the Quality of the Weak Annotator
- Section 4: new small subsection: 4.4. From Modifying the Learning Rate to Weighted Sampling
- Section 5: adding some related works
- Appendix A: more explanation and some experiments backing up the rationale behind clustered GP

---

### Decision · Program_Chairs · 2018-01-29
**ICLR 2018 Conference Acceptance Decision**

**Decision:**

Accept (Poster)

**Comment:**

This paper introduces a student-teacher method for learning from labels of varying quality (i.e. varying fidelity data). This is an interesting idea which shows promising results.

Some further connections to various kinds of semi-supervised and multi-fidelity learning would strengthen the paper, although understandably it is not easy to cover the vast literature, which also spans different scientific domains. One reviewer had a concern about some design decisions that seemed ad-hoc, but at least the authors have intuitively and experimentally justified them.